# Endothelial Colony-Forming Cells Dysfunctions Are Associated with Arterial Hypertension in a Rat Model of Intrauterine Growth Restriction

**DOI:** 10.3390/ijms221810159

**Published:** 2021-09-21

**Authors:** Stephanie Simoncini, Hanna Coppola, Angela Rocca, Isaline Bachmann, Estelle Guillot, Leila Zippo, Françoise Dignat-George, Florence Sabatier, Romain Bedel, Anne Wilson, Nathalie Rosenblatt-Velin, Jean-Baptiste Armengaud, Steeve Menétrey, Anne-Christine Peyter, Umberto Simeoni, Catherine Yzydorczyk

**Affiliations:** 1Aix Marseille Univ, Institut National de la Santé Et de la Recherche Médicale (INSERM), Institut National de Recherche pour l’Agriculture, l’Alimentation et l’Environnement (INRAe), Center from Cardiovascular and Nutrition research (C2VN), UMR-S 1263, UFR de Pharmacie, Campus Santé, 13385 Marseille, France; Stephanie.SIMONCINI@univ-amu.fr (S.S.); francoise.dignat-george@univ-amu.fr (F.D.-G.); florence.SABATIER-MALATERRE@univ-amu.fr (F.S.); 2Department Woman-Mother-Child, Division of pediatrics, DOHaD Laboratory, Lausanne University Hospital and University of Lausanne, 1011 Lausanne, Switzerland; hanna.coppola@unil.ch (H.C.); angela.rocca@unil.ch (A.R.); isaline.bachmann@unil.ch (I.B.); estelle.guillot@unil.ch (E.G.); Leila.Zippo1@eduvaud.ch (L.Z.); jean-baptiste.armengaud@chuv.ch (J.-B.A.); umberto.simeoni@chuv.ch (U.S.); 3Flow Cytometry Facility, Department of Formation and Research, University of Lausanne, 1011 Lausanne, Switzerland; romain.bedel@unil.ch (R.B.); anne.wilson@unil.ch (A.W.); 4Department of Oncology, University of Lausanne, 1011 Lausanne, Switzerland; 5Department Heart-Vessels, Division of Angiology, Lausanne University Hospital and University of Lausanne, 1011 Lausanne, Switzerland; nathalie.rosenblatt@chuv.ch; 6Department Woman-Mother-Child, Neonatal Research Laboratory, Clinic of Neonatology, Lausanne University Hospital and University of Lausanne, 1011 Lausanne, Switzerland; Steeve.Menetrey@chuv.ch (S.M.); Anne-Christine.Peyter@chuv.ch (A.-C.P.)

**Keywords:** intrauterine growth restriction, developmental programming, arterial hypertension, endothelial colony-forming cells, oxidative stress, stress-induced premature senescence

## Abstract

Infants born after intrauterine growth restriction (IUGR) are at risk of developing arterial hypertension at adulthood. The endothelium plays a major role in the pathogenesis of hypertension. Endothelial colony-forming cells (ECFCs), critical circulating components of the endothelium, are involved in vasculo-and angiogenesis and in endothelium repair. We previously described impaired functionality of ECFCs in cord blood of low-birth-weight newborns. However, whether early ECFC alterations persist thereafter and could be associated with hypertension in individuals born after IUGR remains unknown. A rat model of IUGR was induced by a maternal low-protein diet during gestation versus a control (CTRL) diet. In six-month-old offspring, only IUGR males have increased systolic blood pressure (tail-cuff plethysmography) and microvascular rarefaction (immunofluorescence). ECFCs isolated from bone marrow of IUGR versus CTRL males displayed a decreased proportion of CD31+ versus CD146+ staining on CD45− cells, CD34 expression (flow cytometry, immunofluorescence), reduced proliferation (BrdU incorporation), and an impaired capacity to form capillary-like structures (Matrigel test), associated with an impaired angiogenic profile (immunofluorescence). These dysfunctions were associated with oxidative stress (increased superoxide anion levels (fluorescent dye), decreased superoxide dismutase protein expression, increased DNA damage (immunofluorescence), and stress-induced premature senescence (SIPS; increased beta-galactosidase activity, increased p16^INK4a^, and decreased sirtuin-1 protein expression). This study demonstrated an impaired functionality of ECFCs at adulthood associated with arterial hypertension in individuals born after IUGR.

## 1. Introduction

Subjects born after IUGR are at an increased risk of higher blood pressure during infancy [1], adolescence [2,3], young adulthood [4], and later in life [5,6,7]. Among the mechanisms potentially involved in the developmental programming of hypertension, alterations of the vascular system have been shown to play an important role in addition to the long-term effects of a decreased nephron number endowment and hypothalamic-pituitary-adrenal axis hyperactivity [8]. Low-birth-weight subjects display increased arterial stiffness, increased intima-media thickness, decreased arterial compliance, and impaired endothelium-dependent vasodilation [9,10,11,12]. The endothelium is considered a dynamic organ with different functions, which together regulate the antithrombotic and anti-inflammatory states, improve angiogenesis, and regulate vascular tone and tissue perfusion. Endothelial progenitor cells (EPCs) are critical circulating components of the endothelium, and are identified as key factors in endothelial repair. EPCs can be distinguished according to their phenotype and functional properties. Early EPCs are of hematopoietic origin and promote angiogenesis through paracrine mechanisms, but cannot give rise to mature endothelial cells [13,14,15,16]. In contrast, endothelial colony-forming cells (ECFCs) or late outgrowth EPCs [13] have clonal potential and the capacity to produce mature endothelial cells and promote vascular formation in vitro and in vivo. In particular, these cells are able to proliferate, auto-renew, migrate, differentiate, and promote vascular growth and neovascularization. In the clinical setting, decreased numbers and altered functionality of EPCs have been observed in various cardiovascular disorders. In adult patients, reduced numbers of circulating EPCs have been associated with cardiovascular disease [17] and have been inversely correlated with arterial blood pressure values. In newborns, a positive correlation was observed between birth weight and the number of circulating endothelial progenitor cells [18]. ECFCs from low-birth-weight and preterm infants displayed reduced numbers and dysfunction [19]. Ligi et al. showed impaired angiogenic properties in ECFCs from low-birth-weight neonates [20,21]. Several factors regulating EPC number and functionality have been identified. Notably, it has been reported that oxidative stress and cellular senescence can negatively modulate EPC number and functionality [22]. Satoh et al. observed increased oxidative DNA damage in EPCs from adult patients with coronary artery disease [23]. In animal models, increased ROS production has been associated with reduced EPC mobilization in bone marrow in the early post-infarction phase [24]. Imanishi et al. showed that beta-galactosidase activity was increased and telomerase activity decreased in EPCs isolated from patients with hypertension, and that the induction of cellular senescence was due to angiotensin-II-mediated oxidative stress [25]. Oxidative stress and cellular senescence have been associated with fetal growth restriction [26,27]. ECFCs from preterm infants have an increased vulnerability to hyperoxia-induced oxidative stress leading to cell dysfunction [28,29]. In pregnant women with growth-restricted fetuses, increased malondialdehyde [30], increased urinary 8-oxo-7,8 dihydro-2′deoxyguanosine, increased plasma protein carbonylation, and decreased total antioxidant capacity have been observed [31], all of which are consistent with similar observations made in IUGR neonates [31,32,33]. However, the relationship between altered endothelial function in IUGR subjects and oxidative stress is not completely understood. In addition, Ligi et al. observed impaired proliferation, vascular network formation, and angiogenic capabilities of ECFCs isolated from the cord blood of low-birth-weight newborns, associated with accelerated senescence [20,21,34]. However, it is not well identified whether these early dysfunctions of ECFCs persist through adulthood and constitute a possible link between IUGR and arterial hypertension development, and which mechanisms could be involved.

Using a recognized rat model of developmental programming of arterial hypertension related to IUGR, we investigated whether the proportion of ECFCs, their proliferative capacity, and their vascular network formation are altered. We also studied their angiogenic capacity and explored some markers related to oxidative stress and accelerated senescence.

## 2. Results

### 2.1. IUGR-Induced Lower Body Weight at Birth and at Six Months of Life

We observed a significantly decreased body weight at birth in both sexes in the IUGR group compared with the CTRL group (−32% for males and −33% for females). This decrease in weight persisted at 6 months of life in the IUGR group (−22% for males and −15% for females).

### 2.2. IUGR-Induced Increased Systolic Blood Pressure and Microvascular Rarefaction

The systolic blood pressure (SBP) was assessed using the tail-cuff method in six-month-old rats. We observed an increased SBP in IUGR males (+22%; *p* < 0.01) compared with CTRL, but no difference in females (Table 1). The capillary density was evaluated using lectin-TRITC staining. We observed a reduction in lectin staining (−58%; *p* < 0.01) in the tibial muscles of IUGR vs. CTRL males (Figure 1A). No difference was observed between CTRL and IUGR females (Figure 1B).

### 2.3. Decreased Number of IUGR-ECFCs

Using flow cytometry, we observed a decrease (−50%; *p* < 0.05) in the proportion of CD31+ versus CD146+ staining on CD45-viable cells IUGR-ECFCs compared with CTRL-ECFCs (Figure 2A–D). Moreover, CD34 was significantly less expressed (−50%; *p* < 0.05) in IUGR-ECFCs vs. CTRL-ECFCs (Figure 3A). The number of ECFCs can be modulated by inflammatory parameters. Therefore, we measured interleukin-1 beta (IL-1β) expression by western blot in adipose tissue, which is considered the major source of pro-inflammatory cytokine secretion. In IUGR males IL-1β expression was significantly higher (+110%; *p* < 0.05) compared with CTRL males (Figure 3C).

### 2.4. Altered Proliferation and Capillary-like Outgrowth Sprout Properties

The proliferative capacity of ECFCs in both groups was assessed by measuring absorbance at 450 nm at 6 and 24 h after BrdU incorporation. Compared with CTRL-ECFCs, we observed a significantly reduced proliferation capacity in IUGR-ECFCs at 6 h (−48%; *p* < 0.01) and at 24 h (−75%; *p* < 0.01) (Figure 4A). We also evaluated the capacity of ECFCs to form a capillary-like structure at 6 and 24 h using Matrigel cultures and observed a greatly altered capacity at both time points in IUGR-ECFCs vs. CTRL-ECFCs. Indeed, they formed open, short capillary-like structure tubes with a reduction in the number of closed tubes and branches (Figure 4B).

### 2.5. Impaired Angiogenic Capacity

NO production in ECFCs was assessed with fluorescent DAF-2DA. In IUGR-ECFCs compared with CTRL-ECFCs, we observed a decreased basal NO production (−41%; *p* < 0.01) (Figure 5A) as well as after stimulation by acetylcholine (−56%; *p* < 0.01) (Figure 5B). We also observed decreased eNOS protein expression in IUGR-ECFCs compared with CTRL-ECFCs by immunofluorescence (−66%; *p* < 0.05) and by western blot (−41%; *p* < 0.05) (Figure 6A,C).

We evaluated the angiogenic profile of ECFCs by immunofluorescence. In IUGR-ECFCs vs. CTRL-ECFCs, we observed a decrease in expression of angiopoietin (−48%; *p* < 0.05) (Figure 7A), angiomotin (−31%; *p* < 0.05) (Figure 7B), vascular endothelial growth factor receptor-2 (VEGFR-2) (−42%; *p* < 0.05) (Figure 7C), and vascular endothelial growth factor-A (VEGF-A) (−41%; *p* < 0.05) (Figure 7D). However, we observed a slight increase in the expression of thrombospondin-1 (+77%; *p* < 0.05) (Figure 7E).

### 2.6. Oxidative Stress

We measured superoxide anion production using the oxidative fluorescent dye hydroethidine and observed an increase in superoxide anion production (+511%; *p* < 0.001) in IUGR-ECFCs vs. CTRL-ECFCs (Figure 8A). We determined the source of superoxide anion production using pre-incubation with apocynin and L-NAME, inhibitors of NADPH oxidase and eNOS, respectively. In IUGR-ECFCs, superoxide anion production was significantly decreased after treatment with apocynin (−81%; *p* < 0.001) or L-NAME (−77%; *p* < 0.05) (Figure 8B). In contrast, no effect on superoxide anion production was observed in CTRL-ECFCs after treatment with these two inhibitors (Figure 8C).

We measured the protein expression of Cu/Zn superoxide dismutase and catalase in CTRL-ECFCs and IUGR-ECFCs by western blot. In IUGR-ECFCs vs. CTRL-ECFCs, we observed a decrease in expression of Cu/Zn superoxide dismutase (−27%; *p* < 0.05), but no difference in the expression of catalase (Figure 9A). In addition, we measured 53BP-1 expression, a well-known DNA damage response factor, and observed an increase in 53BP-1 staining (+204%; *p* < 0.05) in IUGR-ECFCs compared with those from the CTRL group (Figure 9B).

### 2.7. Cellular Senescence

Cellular senescence was evaluated by measurement of senescence-associated–beta-galactosidase (SA-β-gal). We observed an increase in beta-galactosidase staining (+103%; *p* < 0.01) in IUGR-ECFCs compared with CTRL-ECFCs (Figure 10). We measured the protein content of some senescence markers such as sirtuin-1, p21^WAF^, and p16^INK4a^, and observed a decrease in sirtuin-1 protein expression by immunofluorescence (−63%; *p* < 0.05) (Figure 11A) and by western blot (−31%; *p* < 0.05) (Figure 11B). No difference in p21^WAF^ expression between IUGR-ECFCs and CTRL-ECFCs (Figure 12A) was noted. However, we observed an increase (+141%; *p* < 0.05) in p16^INK4a^ protein expression in IUGR-ECFCs (Figure 12B). We also found an increased phosphorylated p38 MAPK^Thr180+Tyr182^/ p38MAPK protein expression in IUGR-ECFCs (+62%) (Figure 12C,D).

## 3. Discussion

The findings from this study demonstrate that IUGR induced an increase in SBP and the presence of microvascular rarefaction only in in six-month-old male rats. These changes were associated with a reduction in the proportion of CD31+ versus CD146+ staining on CD45-cells and CD34 expression in ECFCs and an alteration in their functionality. This is shown by reduced proliferation and impaired capillary-like structure formation capability, as well as altered eNOS expression and angiogenic profiles. These dysfunctions are related to oxidative stress and stress-induced premature cellular senescence (SIPS).

Prenatal exposure to maternal undernutrition in rats is well known to induce IUGR in offspring, and is characterized by a low birth weight and associated with the development of cardiometabolic disorders at adulthood [35,36].

We observed in our rat model that 9% casein administrated to dams throughout gestation led to a reduction of body weight in both sexes at birth and at six months of life. However, only six-month-old male rats displayed an increase in SBP at this age. In fetal programing of arterial hypertension in rats, it is well established that after puberty only male growth-restricted offspring remain hypertensive. In contrast, female growth-restricted offspring stabilize their blood pressure to the level of adult female controls. It has been suggested that estrogen contributes to the normalization of arterial blood pressure in female growth-restricted offspring at adulthood [37].

Reduced density of arterioles and capillaries, also named microvascular rarefaction, is a mechanism that increases peripheral vascular resistance and contributes to the pathophysiology of arterial hypertension [38] in both animal and human studies [39]. We observed reduced capillary and arteriole density in IUGR compared with CTRL males at six months of life, as observed in another rat model of developmental programing of arterial hypertension [40]. Reduced microvascular density has been related to impaired angiogenesis, as demonstrated on aortic rings of a similar rat model of IUGR [41].

Angiogenesis is important to maintain the integrity of tissue perfusion, which is crucial for physiologic organ function, and so impaired angiogenesis could contribute to the development of hypertension. EPCs, and more particularly ECFCs, play a major role in the angiogenic process by maintaining microvasculature and stimulating postnatal angiogenesis [42]. As we observed increased SBP and microvascular rarefaction only in IUGR males, in this study we explored ECFC functionality in CTRL and IUGR males.

Previous studies have demonstrated that ECFCs express surface markers such as CD31, CD146, and CD34 and are negative for CD45 [43,44]. Using flow cytometry, we observed a reduced proportion of CD31+ versus CD146+ staining on CD45-cells in IUGR-ECFCs compared with CTRL-ECFCs. In addition, we used immunofluorescence to measure CD34 expression, a marker of tube-forming capacity [45], a property related to ECFCs, and observed decreased CD34 expression in IUGR-ECFCs compared with CTRL-ECFCs. A negative correlation between circulating EPC numbers and multiple cardiovascular risks at adulthood has been observed in several human studies [46,47,48,49]. However, in individuals born with low birth weight, data are relatively scarce. A smaller number of circulating EPCs isolated from cord blood at birth has been observed in preterm infants [50] as well as in IUGR-complicated pregnancies [51]. In addition, Meister et al. observed a decrease of 50% in CD34+ cells in preterm neonates compared with term neonates [52]. These data suggest that deleterious conditions during fetal life could alter EPC numbers. Particularly, inflammation has been associated with IUGR as inflammatory cytokines seem to play an important role in this process. Indeed, increased levels of pro-inflammatory cytokines such as interleukin-8, interferon-gamma, and tumor necrosis factor-alpha have been observed in individuals born with fetal growth restriction [53]. In addition, increased IL-1β levels have been observed in newborns with low birth weight [54]. Moreover, high levels of IL-1β and tumor necrosis factor-alpha negatively modulate CD34 expression at the antigen as well as the mRNA levels [55]. Adipose tissue is the major source of proinflammatory cytokine production. In adipose tissue from the same group of animals from which ECFCs were isolated, we observed that IUGR males displayed a significant increase in IL-1β protein expression compared with CTRL males. This suggests that inflammation could play a role in the decreased proportion of IUGR-ECFCs; however, the mechanisms involved are still unknown.

We explored the functionality of ECFCs by measuring both DNA synthesis and their capacity to form a capillary-like structure. We observed decreased proliferative capability at 6 h, which persisted at 24 h, and reduced capillary-like structure formation at both 6 and 24 h in IUGR-ECFCs vs. CTRL-ECFCs, suggesting that IUGR-ECFCs have a reduced ability to migrate and thus repair vascular damage. Ligi et al. demonstrated that ECFCs isolated from cord blood of low-birth-weight neonates displayed reduced proliferation capability and capillary-like structure formation [21]. To our knowledge, this study is the first to observe that these early dysfunctions of ECFCs persist thereafter at adulthood. The impaired functionality of IUGR-ECFCs could be due to impaired angiogenesis. Nitric oxide (NO) is necessary for angiogenesis to occur [56], is involved in the mobilization of EPCs, and improves their migratory and proliferative activities [57], notably by regulation of their angiogenic activity [58]. We observed decreased NO production in IUGR-ECFCs compared with CTRL-ECFCs under basal conditions as well as after stimulation by acetylcholine. Decreased NO production has been related to endothelial dysfunction and arterial hypertension, and is often observed in individuals born after IUGR [8,59]. Moreover, a link between eNOS expression/functionality and EPC function has also been described [60]. This could explain the reduced proliferative and migration capabilities of IUGR-ECFCs. Similar observations have been made in human umbilical vein endothelial cells exposed to hypoxia [61], a condition strongly associated with IUGR [62]. NO can interact with angiogenic factors. Vascular endothelial growth factor (VEGF) plays an important role in EPC differentiation and vascular repair [63,64]. A reciprocal relation between NO and VEGF has been demonstrated as synthesis of VEGF can be induced by NO [65,66], and VEGF increases NO production by eNOS promoting angiogenesis [67]. Thus reduced NO bioavailability in IUGR-ECFCs could have an impact on VEGF expression and subsequently on ECFC functionality, as observed in patients with coronary heart disease [68]. VEGF-A is the most important VEGF family member and was the first to be characterized [69]. Three receptors have been identified, VEGFR-1 (Flt-1), VEGFR-2 (Flk-1), and VEGFR-3 (Flt-3). Amongst them, VEGFR-2 is able to bind to VEGF-A with an affinity 10-fold lower than that of VEGFR-1. However VEGFR-2 is the main mediator of VEGF-A activity in endothelial cell functions such as differentiation, proliferation, migration, angiogenesis, and vessel permeabilization [70]. We observed reduced VEGF-A and VEGFR-2 expression in IUGR-ECFCs. Decreased VEGF-A expression has been observed in preeclamptic pregnancy, an obstetric complication often associated with an increased incidence of IUGR [71], and in the pancreatic islets of IUGR fetal sheep [72]. In addition, VEGF was markedly downregulated in EPCs isolated from patients with either coronary heart disease [73] or from diabetic patients, and is associated with decreased eNOS expression [74]. Ligi et al. observed that VEGF-A expression was significantly decreased in ECFCs from low-birth-weight newborns [20].

NO and VEGF also interact with other angiogenic factors such as angiopoietin-1, angiomotin, and thrombospondin. In EPCs, angiopoietin-1 regulates their mobilization from the bone marrow [75] and improves neovascularization thanks to NO [76,77]. Angiopoietin-1 alone does not stimulate proliferation and tube formation of endothelial cells in vitro, but seems to interact downstream with VEGF [78]. Angiomotin plays an important role in proliferation and function of endothelial cells, as well as in the regulation of tube formation [79]. On the contrary, thrombospondin-1 inhibits the migration of endothelial cells and tubule formation in ECFCs, as well as VEGF release from the extracellular matrix and VEGF signal transduction. We observed a decreased expression of angiopoietin-1 and angiomotin in IUGR-ECFCs, as reported in pregnancies complicated by preeclampsia with IUGR [80], in endothelial cells from knockdown angiomotin zebrafish [81], and also during the postnatal period in ECFCs from low-birth-weight newborns [80].

In contrast, we observed an increased expression of thrombospondin-1 in IUGR-ECFCs. In EPCs, an up-regulation of thrombospondin-1 mRNA expression related to impaired reendothelialization function in vitro and in vivo has been observed in diabetic patients [74], and in ECFCs isolated from low-birth-weight newborns [20].

These data suggest that IUGR-ECFCs display an imbalance in their angiogenic profile. An up-regulation of this anti-angiogenic factor could be related to the reduced proliferation and impaired capillary-like outgrowth sprout formation capability that is observed in the IUGR group.

Several factors have been shown to negatively regulate EPC functionality, such as oxidative stress and cellular senescence. Oxidative stress occurs when the amount of reactive oxygen species (ROS) exceeds the antioxidant capacity, resulting in an imbalance between ROS production and elimination. ROS are chemically reactive components formed during the metabolism of oxygen molecules and are mainly produced in endothelial cells by NADPH oxidase and eNOS uncoupling [82,83]. Excessive ROS can react with cellular macromolecules leading to altered biological activity. ECFCs are highly sensitive to oxidative stress [84,85], and so to improve tissue repair they must have an antioxidant defense system to survive. Compared with CTRL-ECFCs, we observed an increase in superoxide anion production in IUGR-ECFCs, which was mediated by NADPH oxidase and eNOS uncoupling as identified using apocynin and L-NAME, inhibitors of NADPH oxidase and eNOS, respectively. These inhibitors have no effect on superoxide anion production in CTRL-ECFCs. Other animal models of hypertension have also displayed increased NADPH-driven superoxide generation in vessels [86,87,88]. In IUGR-ECFCs, we observed decreased Cu/Zn superoxide dismutase but no difference in catalase expression, which could explain the accumulation of the superoxide anion because Cu/Zn SOD cannot correctly catalyze the dismutation of superoxide to hydrogen peroxide and O_2_. Oxidative stress can induce lipid, protein, and DNA damage. In IUGR-ECFCs, we observed increased 53BP-1 staining, which is an important regulator of the cellular response to DNA double-stranded break repair that promotes the end-joining of distal DNA ends [89]. In addition, hyperactivity of the renin-angiotensin system has been associated with altered number and functionality of EPCs [90]. We have not explored the activity of the renin-angiotensin system in ECFCs, but in a similar animal model of IUGR we previously showed an upregulation of this system in carotid arteries, characterized by an exaggerated response to angiotensin II and increased expression of the angiotensin II receptor type-1 [90,91].

Excessive ROS levels and the presence of DNA damage can contribute to cellular senescence of endothelial cells, notably by decreasing NO production and impaired angiogenesis thus altering vascular repair [92]. We evaluated cellular senescence by measurement of SA-β-gal activity, which is the most extensively used biomarker [93]. We observed an increase in activity of SA-β-gal in IUGR-ECFCs compared with CTRL-ECFCs, as observed in early EPCs isolated from a rat model of IUGR [94] and ECFCs isolated from low-birth-weight newborns [34].

In addition, we also measured some factors related to cellular senescence such as p21^WAF^, p16^INK4a^, and sirtuin-1, a NAD+ deacetylase [89]. In IUGR-ECFCs compared with CTRL-ECFCs, we observed no difference in p21^WAF^ expression, but increased p16^INK4a^ protein expression and a decrease in expression of the anti-aging protein sirtuin-1, suggesting the presence of SIPS, which could be reversed in contrast to replicative senescence. Thus, SIPS could be associated with the impaired functionality observed in IUGR-ECFCs. In low-birth-weight newborns, Vassallo et al. showed that SIPS was associated with impaired proliferation, capillary-like structure formation, and angiogenic factors [34].

These dysfunctions of IUGR-ECFCs related to SIPS might be due to an altered secretory phenotype, named senescence-associated secretory phenotype (SASP), characterized by secretion of growth factors, proteases, inflammatory cytokines, and the release of extracellular vesicles [95,96]. Notably, as we found no difference in p21^WAF^ expression, the DNA damage observed in IUGR-ECFCs could be due to the presence of SASP. Indeed, SASP could contribute to the impaired functional properties of cells and tissues and so promote the progression of aging-related diseases [97]. Activation of p38MAPK activity [98] is associated with SASP [99], and is a major signaling pathway regulating DNA damage and senescence in response to oxidative stress [100], as demonstrated by Shen et al. on human umbilical vein endothelial cells [101]. Increased activity of p38MAPK protein characterized by an increased ratio of phosphorylated p38MAPK^Thr180+Tyr182^/p38MAPK has been observed in IUGR-ECFCs. In addition, SIPS could be related to increased IL-1β expression, as observed in adipose tissue from IUGR males, and as found in the cellular senescence of human umbilical vein endothelial cells [102]. These data suggest that SIPS might be induced by SASP and could be associated with the impaired functions observed in IUGR-ECFCs.

## 4. Materials and Methods

### 4.1. Body Weight Measurement

The body weight was assessed at birth and at six months of life in both groups and both sexes.

### 4.2. Animal Model

We used a rat model of IUGR (Swiss Veterinarian Animal Care committee-VD3050-31.01.2017) [103]. Pregnant rats were randomly allocated during gestation to a control diet (23% casein (version 0001 210 SAFE, Augy, France); CTRL group) or to an isocaloric low-protein diet (9% casein (version 0040), SAFE); IUGR group). Each litter was then equalized to ten pups per group to ensure a standardized nutrient supply until weaning, and thereafter rats from both groups had free access to a standard diet (A04, SAFE Diets, Augy, France) and water. Every sample animal originated from a separate litter.

### 4.3. Systolic Blood Pressure Measurement

SBP was measured in both sexes in CTRL (*n* = 5) and in IUGR (*n* = 5) rats as previously described [59]. Briefly, the SBP was measured in six-month-old conscious animals using the tail-cuff plethysmography method associated with thermostatically warmed restrainers designed for rodents and adapted to the size of the animal (CODA™ High Throughput System-Kent Scientific Corporation, Torrington, CT, USA). Each animal was acclimatized to this procedure during one week before measurements, which were always performed by a single operator.

### 4.4. Microvascular Density Measurement

Morphological measurements of microvascular density were performed in both sexes on anterior tibialis muscle sections from six-month-old CTRL (*n* = 5) and IUGR (*n* = 5) male rats using lectin-tetramethylrhodamine (TRITC) (Sigma-Aldrich, Saint Louis, MO, USA), as previously described [41]. Briefly, anterior tibialis muscle sections were stained with lectin-TRITC (1/100) overnight at 4 °C then were rinsed with phosphate-buffered saline (PBS) and mounted using Fluoromount-G medium with 4′6-diamidino-2-phenylindole (DAPI; Interchim, France). The slides were observed blindly by the same experimenter using a fluorescence microscope (Eclipse Ti2 Series-Nikon Europe B.V, Amsterdam, the Netherlands). Fluorescence of lectin was normalized to DAPI fluorescence and autofluorescence was subtracted. The pictures were evaluated using ImageJ software (Java 1.8.0_112, National Institutes of Health, Southern Montgomery, USA, access on 01 july 2021). Each experiment was performed in duplicate.

### 4.5. Endothelial Progenitor Cell Isolation

Bone marrow was collected from the tibialis and femur of CTRL and IUGR male rats at six months of life. Briefly, bone marrow mononuclear cells were isolated by density gradient centrifugation by diluting 1:1 in PBS and layering over a separating medium (Histopaque 1077-, Sigma-Aldrich, Saint Louis, MO, USA). After 30 min centrifugation at 400× *g*, mononuclear cells isolated from the interface were washed three times in Roswell Park Memorial Institute medium (RPMI) medium, 10% fetal calf serum (Thermo Fisher Scientific, Rockford, IL, USA), and resuspended in endothelial basal cell growth culture medium-2 (EBM2) supplemented with endothelial cell growth medium MV2 (PromoCell, Heidelberg, Germany) and penicillin/streptomycin. ECFCs colonies were identified as well-circumscribed monolayers of cobblestone-appearing cells using an inverted microscope (Nikon, Eclipse Ti2 Series) as previously described [21]. Colonies without cobblestone-like morphology were mechanically removed to prevent them from becoming the predominant cells. The cells were isolated from CTRL (CTRL-ECFCs) and IUGR (IUGR-ECFCs) rats at six-months old and were studied between passages 1–3. The ECFC experiments represent individual animals taken from separate litters. Unfortunately, primary cultures are particularly sensitive, and during the experiments, we had to face contaminations and had to throw away some ECFCs, which explains the difference in number between the experiments.

### 4.6. ECFC Quantification Using Flow Cytometry

Single-cell suspensions from CTRL-ECFCs (*n* = 3) and IUGR-ECFCs (*n* = 5) were stained with fluorochrome-labeled monoclonal antibodies against CD31 PE (TLD-3A12), CD45 FITC (OX-1), and CD146 (LSEC) APC (BD Biosciences, San Jose, CA, USA; or Miltenyi Biotech, Bergisch Gladbach, Germany) in PBS/3%FCS for 20 min at 4 °C. After washing out unbound antibody-conjugates by centrifugation, the cells were resuspended in 200 μL PBS/3%FCS and 0.5 μg DAPI (Thermo Fisher Scientific, Rockford, IL, USA) was added to discriminate dead cells. Samples were analyzed on a LSRII SORP flow cytometer equipped with 5 lasers (BD). Data were analyzed with FlowJo software (FlowJo v10.8, Ashland, OR, USA).

### 4.7. ECFC Proliferation Test

The proliferative capacity of CTRL-ECFCs (*n* = 6) and IUGR-ECFCs (*n* = 7) (20,000 cells/well) was measured by DNA synthesis after 6 and 24 h using a colorimetric cell proliferation ELISA test based on the incorporation of 5′-bromo-2′-deoxyuridine (BrdU) during DNA replication (Roche diagnostics, Basel, Switzerland) as previously described [21]. Each experiment was performed in triplicate.

### 4.8. ECFC Capillary-like Structure Formation

The capillary-like structure formation of CTRL-ECFCs (*n* = 5) and IUGR-ECFCs (*n* = 6) (20,000 cells/well) was evaluated in 96-well plates coated with 50 μL of growth factor reduced Matrigel (BD Biosciences) at 6 and 24 h as previously described [21]. Each experiment was performed in triplicate.

### 4.9. Measurement of NO Production by ECFCs

NO production by ECFCs was detected using the NO-specific fluorescent dye, 4,5-diaminofluorescein diacetate (DAF-2DA) [59]. Briefly, CTRL-ECFCs (*n* = 5) and IUGR-ECFCs (*n* = 5) were loaded with DAF-2DA (10 mM) and incubated in a light-protected humidified chamber at 37 °C for 1 h. ECFCs were then incubated for 1 h at 37 °C in N-2-Hydroxyethylpiperazine-N-2-Ethane Sulfonic Acid (HEPES) buffer alone or with acetylcholine (100 mM) added. Digital images were observed blindly by the same experimenter using a fluorescence microscope (Eclipse Ti2 Series). Three images per culture well (2 wells per ECFCs) were captured. Fluorescence of DAF-2DA was normalized to DAPI fluorescence and autofluorescence was subtracted. The pictures were evaluated with ImageJ software. Each experiment was performed in duplicate.

### 4.10. Measurement of Superoxide Anion Production by ECFCs

Superoxide anion production was evaluated in CTRL-ECFCs (*n* = 5) and IUGR-ECFCs (*n* = 5) using the oxidative fluorescent dye hydroethidine (2 μM, Sigma-Aldrich) [36,59] in the presence or absence of the NOS blocker N-nitro-L-arginine methyl ester (L-NAME; 100 μM; 24 h pre-incubation (Sigma Aldrich) and NADPH blocker apocynin (1 mM; 24 h pre-incubation; Millipore Corporation, Burlington, MA, USA) in comparison to autofluorescence detection. Digital images were observed blindly by the same experimenter using a fluorescence microscope (Eclipse Ti2 Series). Three images per culture well (2 wells per ECFC) were captured. The fluorescence of superoxide anion was normalized to DAPI fluorescence and autofluorescence was subtracted. The pictures were evaluated using ImageJ software. Each experiment was performed in duplicate.

### 4.11. Senescence Detection in ECFCs

SA-β-gal activity was performed in CTRL-ECFCs (*n* = 5) and IUGR-ECFCs (*n* = 5) using a senescence detection kit (Cell Signaling Technology, Danvers, MA, USA) according to the manufacturer’s instructions. SA-β-gal-positive cells were normalized as a percentage of the total number of cells [34]. Each experiment was performed in duplicate.

### 4.12. Immunofluorescence

CTRL-ECFCs (*n* = 4–5) and IUGR-ECFCs (*n* = 4) were fixed using cold ethanol 70% and stained with sirtuin-1, endothelial nitric oxide synthase (eNOS), 53BP-1 (rabbit, 1:100; cell signaling, Danvers, MA, USA), CD34, angiopoietin, angiomotin, VEGF-A, VEGFR-2, and thrombospondin-1 (rabbit, 1:100; Abcam, Cambridge, UK) overnight at 4 °C. ECFCs were then washed with PBS and incubated for 2 h with Alexa Fluor-488 goat anti-rabbit IgG (IgG 1:200), Abcam), and were rinsed with PBS and mounted using Fluoromount-G mounting medium with DAPI. Autofluorescence was subtracted. A negative control was obtained using incubation only with the secondary antibody. The slides were observed blindly using a fluorescence microscope (Eclipse Ti2 Series) by the same experimenter. Three images per culture well (2 wells per ECFCs) were captured. Fluorescence of each factor was normalized to DAPI fluorescence. The pictures were evaluated using ImageJ software. Each experiment was performed in duplicate.

### 4.13. Protein Expression Evaluation Using Western Blotting

Proteins were extracted from CTRL-ECFCs (*n* = 3) and IUGR-ECFCs (*n* = 5) using a lysis buffer (HEPES 50 mM, EDTA 1 mM, EGTA 1 mM, Glycerol 10%, pH 7.4, NaF 50 mM, AEBSF 0.1 mM, Leupeptin 10 μg/mL, Pepstatin 5 μg/mL, Aprotinin 3 μg/mL, Sodium Vanadate 1 mM, CHAPS 20 mM) (Sigma-Aldrich). The cell suspension was left on ice for 5 min and then sonicated. The homogenate was centrifuged for 30 min at 10,000 rpm at 4 °C. The supernatant was retained for protein quantification (Life Technologies Europe B.V, Zug, Switzerland) and western blot analysis. Denatured (NuPAGE sample-reducing agent; 10 min at 70 °C) ECFC proteins (35 μg) from the CTRL and IUGR groups were separated on the same gradient gel (NuPAGE 4–12% Bis-Tris gel, Life Technologies Europe B.V) and transferred overnight at 4 °C (30 V) to Whatman nitrocellulose membranes (Life Technologies Europe B.V). Ponceau staining (Life Technologies Europe B.V) confirmed the presence of proteins on the membranes. All primary antibody incubations were performed in blocking buffer (PBS-Tween 2%-bovine serum albumin (BSA) 3%; AppliChem, Darmstadt, Germany) overnight at 4 °C. Antibodies against eNOS, catalase, Cu/Zn superoxide dismutase, IL-1β, sirtuin-1, p21^WAF^, p16^INK4a^, p38 MAPK, phosphorylated p38MAPK^Thr180+Tyr182^, and beta-actin were purchased and used at the dilutions recommended for immunoblotting (1:1000, Cell Signaling Technology Cell Signaling and Abcam). Incubations with HRP anti-mouse or anti-rabbit secondary antibodies (1/2000; Cell Signaling) were performed for 2 h at room temperature in blocking buffer (PBS-Tween 2%-BSA 3%). The antibodies were visualized using enhanced chemiluminescence western blotting substrate (Life Technologies Europe B.V)). A G-BOX Imaging System (GeneSys, Syngene, Cambridge, UK) was used to detect specific bands, and the optical density of each band was measured using specific software (GeneTools 4.03.05.0, Syngene, Cambridge, UK) [89]. We performed sirtuin-1 and Cu/Zn SOD on the same membrane to reduce biological material consumption.

### 4.14. Statistical Analyses

All data were presented as mean ± SEM. Experimental observations were analyzed using the Mann-Whitney U test. GraphPad Prism 8 (version 8.3.0 (538), La Jolla, CA, USA) was used for statistical analyses and creating graphics. The significance level was set at *p* < 0.05.

## 5. Conclusions

### 5.1. Conclusion

The present study demonstrated that at 6 months after birth, adult male rats born after IUGR had a reduced proportion of CD31+ versus CD146+ staining on CD45− cells and CD34 expression in ECFCs, with altered functions of proliferation and capillary-like structure formation. In addition, an imbalance in their angiogenic profile related to oxidative stress and SIPS was observed. These dysfunctions were associated with arterial hypertension and microvascular rarefaction (Figure 13).

### 5.2. Limitations

The present study was performed only in six-month-old male rats. Therefore, it was not possible to determine whether the observed ECFC alterations precede the increase in SBP. To answer this question, it will be necessary to explore the functionality of ECFCs at a younger age at which SBP is not increased. In addition, ECFCs isolated from six-month-old females were not investigated in this study because of the absence of an increase in SBP in these individuals. It will be therefore necessary to determine whether ECFCs are also altered in females even in the absence of increased SBP.

Finally, the present data were obtained in a rat model of IUGR induced by a maternal low-protein diet. Further investigation should be therefore performed in humans in order to determine whether similar alterations could be observed.

### 5.3. Perspectives

In this study, the identification of mechanisms related to ECFC dysfunction, such as oxidative stress and SIPS, could enable us to design specific therapeutic or preventive strategies and to accelerate the research for vascular regenerative therapies. Particularly, it would be interesting to explore whether an antioxidant therapy could restore the functional properties of IUGR-ECFCs, such as proliferation, capillary-like structure formation, and expression of angiogenic factors, associated with a decrease in oxidative stress and reversion of SIPS. Resveratrol is widely known as a phenolic compound with powerful antioxidant activity. Resveratrol is present in several plants, including grape skins, grape seeds, giant knotweed, cassia seeds, passion fruit, white tea, plums, and peanuts [104,105]. Wang et al. demonstrated that resveratrol promotes the proliferation, adhesion, and migration of EPCs in a dose-and time-dependent manner and increases the expression of VEGF to further induce vasculogenesis [106,107], which was mediated by the activation of sirtuin-1 [108]. Resveratrol also delays the senescence of EPCs by increasing telomerase activity to maintain the appropriate levels and function of EPCs [109,110], and by increasing sirtuin-1 functionality [34]. Resveratrol also prevents oxidative stress induced by diabetes in EPCs via sirtuin-1 activation [111]. In ECFCs isolated from low-birth-weight newborns, in vitro treatment with resveratrol has improved ECFC functionality and reversed SIPS; however, whether resveratrol could exert similar actions on ECFCs-IUGR isolated 6 months after birth is still unknown.

In addition, as mentioned above, it will be interesting to explore if impaired ECFC functionalities precede, or are rather a consequence of arterial hypertension by exploring the functionality of ECFCs at birth and at a younger age when SBP is not increased. In addition, although females did not have increased SBP at six months of life, it will be interesting to investigate their ECFC functionality. If ECFC alterations are observed in 6-month-old females, it would be interesting to study whether SBP increases later in life.

Further investigation of epigenetic processes implicated in the regulation of molecular mechanisms identified in this study could be of interest to better understand the developmental programming of hypertension after IUGR.

Because individuals born after IUGR may have subsequent catch-up growth that can amplify cardiometabolic disease, it would also be interesting to observe whether a growth catch-up (induced by litter-size restriction during the lactation period) could amplify the adverse effects related to the IUGR in the present rat model.

Finally, the use of stem cells has emerged as promising for regenerative medicine because of their capacity to contribute to organ repair and regeneration throughout life. In particular, EPCs have been identified as having clinical potential, not only in vascular regenerative applications [112,113] in ischemic diseases such as myocardial infarction and peripheral vascular disease, but also in metabolic diseases and pulmonary and systemic hypertension [114,115]. In particular, ECFCs represent ideal stem cell candidates thanks to their properties of proliferation, autorenewal, migration, differentiation, vascular growth, and neovascularization [116]. Indeed, intrajugular administration of human cord blood-derived ECFCs in newborn rodents was able to reverse alveolar growth arrest, preserve lung vascularity, and reduce pulmonary hypertension in a model of hyperoxia-induced bronchopulmonary dysplasia [42]. This cell therapy also prevented cardiomyocyte hypertrophy, as well as the myocardial and perivascular fibrosis observed after neonatal hyperoxia exposure [117].

Concerning clinical applications, ECFCs could provide an interesting tool in the management of preeclampsia and IUGR and their adverse consequences. Whether ECFC dysfunctions are already present at birth, they could be used as biomarkers to identify individuals with an increased risk to develop cardiometabolic disease later in life and to design specific follow-up or preventative approaches for such individuals. Moreover, identification of mechanisms implicated in ECFC dysfunctions could help to design potential treatments to reverse these alterations, as mentioned above. Such an approach could enable treatment with ECFCs isolated from cord blood before re-injection in the neonate to limit long-term adverse effects of IUGR or preeclampsia. Finally, identification of ECFC dysfunctions in maternal blood in pregnancies complicated by IUGR or preeclampsia could be useful as an early diagnostic tool to predict such complications and to improve their management. This could facilitate the design of therapeutic interventions to limit or prevent the development of IUGR or preeclampsia and thus prevent or limit their adverse consequences. Indeed, preeclampsia often results in IUGR or preterm babies. The level of circulating ECFCs in cord blood of preeclamptic pregnancies was reduced [118,119,120] and impaired angiogenic factors have been associated with preeclampsia. Notably, the angiogenic factor VEGF plays a major role in the management of blood pressure during preeclampsia, and low levels of VEGF have been observed in preeclampsia [121]. Exogenous administration of VEGF has been shown to reverse the antiangiogenic effects of preeclamptic plasma [122], and VEGF represents an important regulator of ECFC functionality. Therefore, based on our present study and these independent observations, future experiments could focus on the “rescue” of ECFC functionality, either by pharmacological treatment or gene therapy, notably by increasing their angiogenic potential by in vitro conditioning (eNOS, VEGF, CD146) as previously published [123,124].

## Figures and Tables

**Figure 1 ijms-22-10159-f001:**
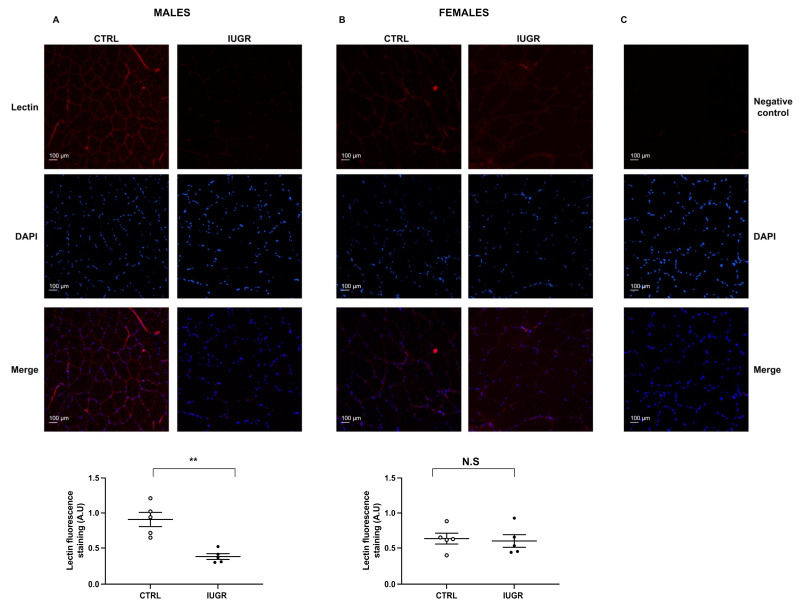
Microvascular density measurement. Capillary density was assessed in the anterior tibialis muscle of CTRL and IUGR males (**A**) and females (**B**) at six months of life using lectin-TRITC staining. Nuclei were counterstained with DAPI, and a negative control was performed (**C**). Magnification (20×); *n* = 5 animals/group; ** *p* < 0.01. N.S: not significant. Scale bar = 100 μm.

**Figure 2 ijms-22-10159-f002:**
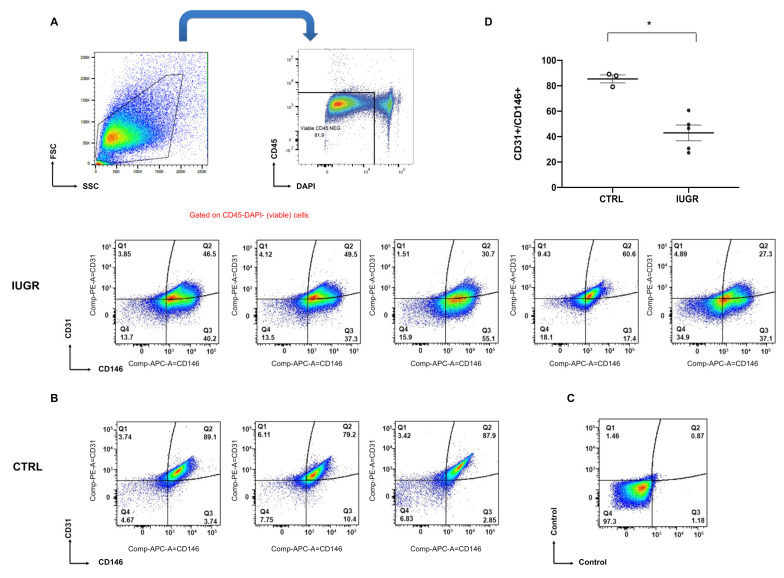
ECFC quantification. Flow cytometry analysis of cultured cells was performed on CTRL-ECFCs and IUGR-ECFCs isolated from six-month-old male rats. (**A**–**D**). Left panel; FSC versus SSC plot. Cells were gated to exclude subcellular debris. Right panel; CD45 versus DAPI staining on gated cells from left panel (**A**). Dead cells (DAPI+) and hematopoietic cells (CD45+) were excluded by gating CD31+ versus CD146+ staining on CD45-viable cells (**B**). Upper panel, cells from IUGR rats, lower-left three panels from CTRL rats, and values were reported in the histogram (**D**). Negative control stain on CD45-viable cells (in the absence of CD31 and CD146) was represented in the lower-right panel (**C**); * *p* < 0.05.

**Figure 3 ijms-22-10159-f003:**
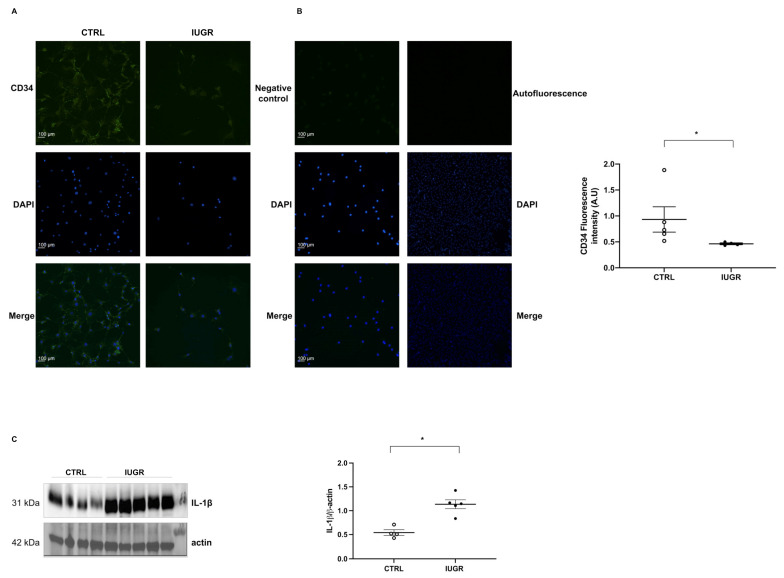
CD34 expression in ECFCs and IL-1β protein expression. CD34 expression was detected by immunostaining in CTRL-ECFCs and IUGR-ECFCs isolated from six-month-old male rats. Magnification (20×). Nuclei were counterstained with DAPI and a negative control (with no primary antibody) and a test for autofluorescence were performed. These pictures are representative images from *n* = 4–5 animals/group. * *p* < 0.05 (**A**). Negative control and autofluorescence tests were performed (**B**). Scale bar = 100 μm. In addition, IL-1β protein expression was measured in adipose tissue from six-month-old CTRL and IUGR male rats (**C**). *n* = 3–5 animals/group. * *p* < 0.05.

**Figure 4 ijms-22-10159-f004:**
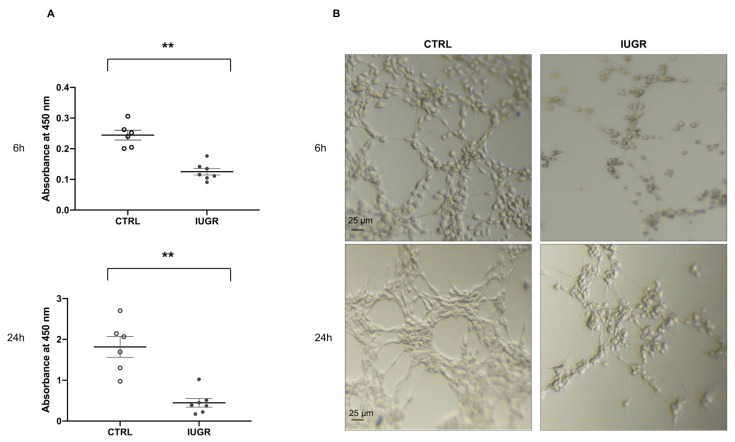
Proliferation and capillary-like structure formation properties of ECFCs. The proliferation capacity of CTRL-ECFCs and IUGR-ECFCs isolated from six-month-old male rats was quantified using BrdU incorporation at 6 and 24 h; *n* = 6–7 animals/group; ** *p* < 0.01 (**A**). The capillary-like outgrowth sprouts were evaluated using Matrigel cultures at 6 and 24 h in CTRL-ECFCs and IUGR-ECFCs isolated from six-month-old male rats. Magnification (5×) (**B**). These pictures are representative images from *n* = 5–6 animals/group. Scale bar = 25 μm.

**Figure 5 ijms-22-10159-f005:**
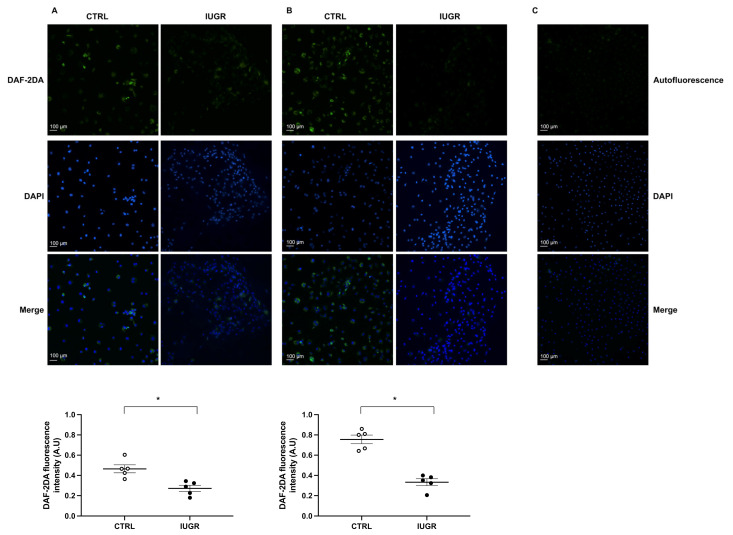
NO production in ECFCs. NO production was evaluated using DAF-2DA in CTRL-ECFCs and IUGR-ECFCs isolated from six-month-old male rats under baseline conditions (**A**) and after stimulation by acetylcholine (**B**). Magnification (20×). Nuclei were counterstained with DAPI. An autofluorescence test was performed (**C**). These pictures are representative images from *n* = 5 animals/group; * *p* < 0.05. Scale bar = 100 μm.

**Figure 6 ijms-22-10159-f006:**
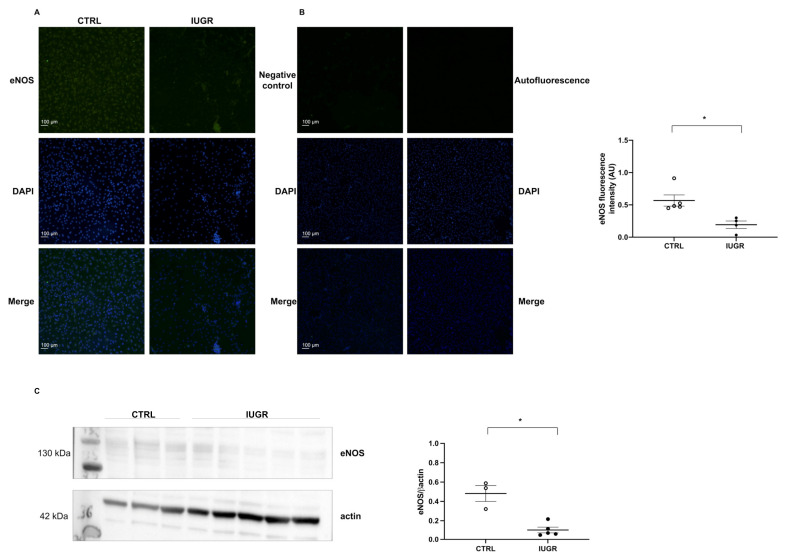
eNOS protein expression in ECFCs. eNOS protein expression was measured by immunofluorescence in CTRL-ECFCs and IUGR-ECFCs isolated from six-month-old male rats (**A**). Magnification (20×). Nuclei were counterstained with DAPI, and a negative control (with no primary antibody) and test for autofluorescence were performed (**B**). These pictures are representative images from *n* = 4–5 animals/group; * *p* < 0.05. Scale bar = 100 μm. eNOS expression was also measured in CTRL-ECFCs and IUGR-ECFCs by western blot (**C**).

**Figure 7 ijms-22-10159-f007:**
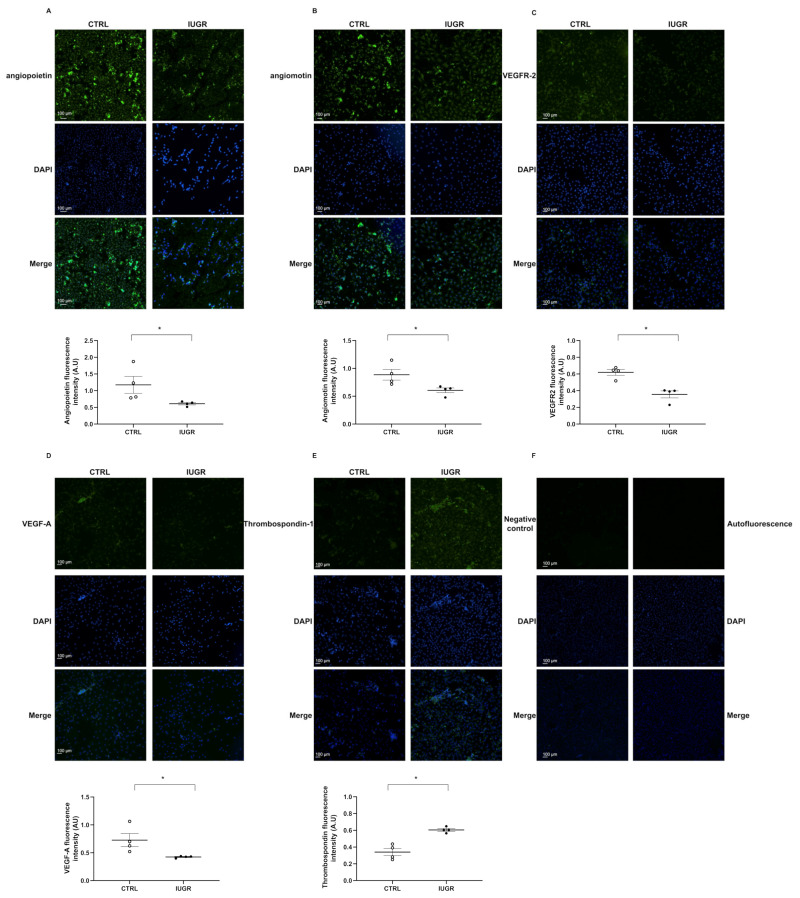
Angiogenic profile of ECFCs. Angiopoietin (**A**), angiomotin (**B**), VEGFR-2 (**C**), VEGF-A (**D**), and thrombospondin-1 (**E**) protein expression was measured by immunofluorescence in CTRL-ECFCs and IUGR-ECFCs isolated from six-month-old male rats. Magnification (20×). Nuclei were counterstained with DAPI. A negative control (with no primary antibody) and test of autofluorescence were performed (**F**). These pictures are representative images from *n* = 4 animals/group; * *p* < 0.05. Scale bar = 100 μm.

**Figure 8 ijms-22-10159-f008:**
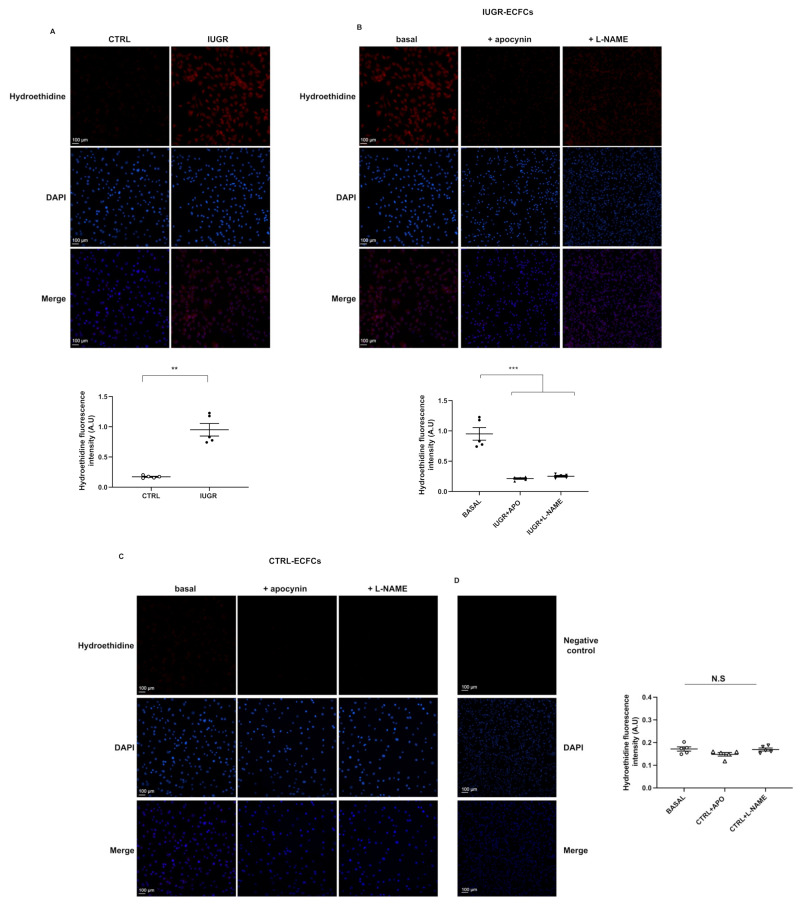
Superoxide anion production in ECFCs. The superoxide anion level was evaluated by hydroethidine in CTRL-ECFCs and IUGR-ECFCs isolated from six-month-old male rats under baseline conditions (**A**) and after 24 h pre-incubation with L-NAME (100 μM) and apocynin (APO; 1 mM) in IUGR-ECFCs (**B**) and CTRL-ECFCs (**C**). Magnification (20×). Nuclei were counterstained with DAPI, and a test of autofluorescence was performed (**D**). These pictures are representative images from *n* = 5 animals/group. ** *p* < 0.01; *** *p* < 0.001; N.S: not significant. Scale bar = 100 μm.

**Figure 9 ijms-22-10159-f009:**
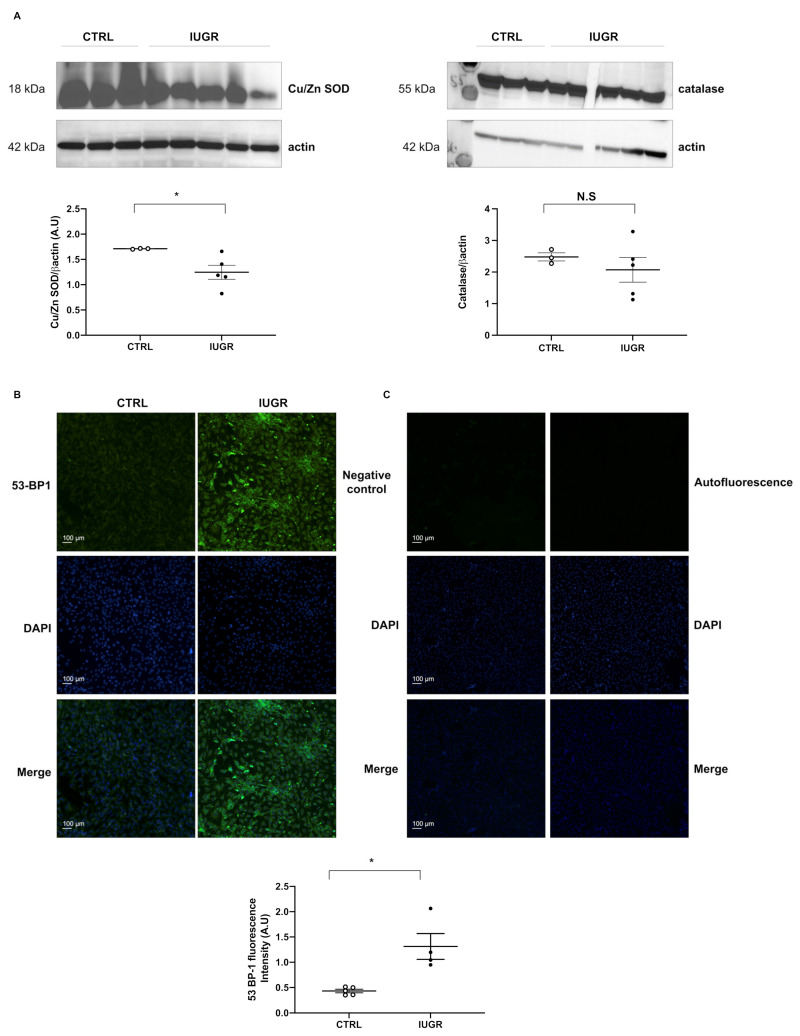
Antioxidant protein expression and DNA damage in ECFCs. The expression of Cu/Zn SOD and catalase proteins was measured by western blot in CTRL-ECFCs and IUGR-ECFCs isolated from six-month-old male rats; *n* = 3–5 animals/group; * *p* < 0.05 (**A**). DNA double-strand breaks were evaluated by 53BP-1 staining in CTRL-ECFCs and IUGR-ECFCs isolated from six-month-old male rats. Magnification (20×) (**B**). Nuclei were counterstained with DAPI, and a negative control (with no primary antibody) and test of autofluorescence were performed (**C**). These pictures are representative images from *n* = 4–5 animals/group; * *p* < 0.05; N.S: not significant. Scale bar = 100 μm.

**Figure 10 ijms-22-10159-f010:**
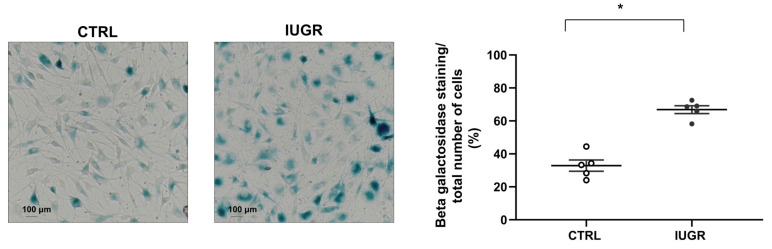
Beta-galactosidase activity in ECFCs. Beta-galactosidase activity was determined as the blue staining normalized to the total number of cells in CTRL-ECFCs and IUGR-ECFCs isolated from six-month-old male rats. Magnification (20×). These pictures are representative images from *n* = 5 animals/group; * *p* < 0.05. Scale bar = 100 μm.

**Figure 11 ijms-22-10159-f011:**
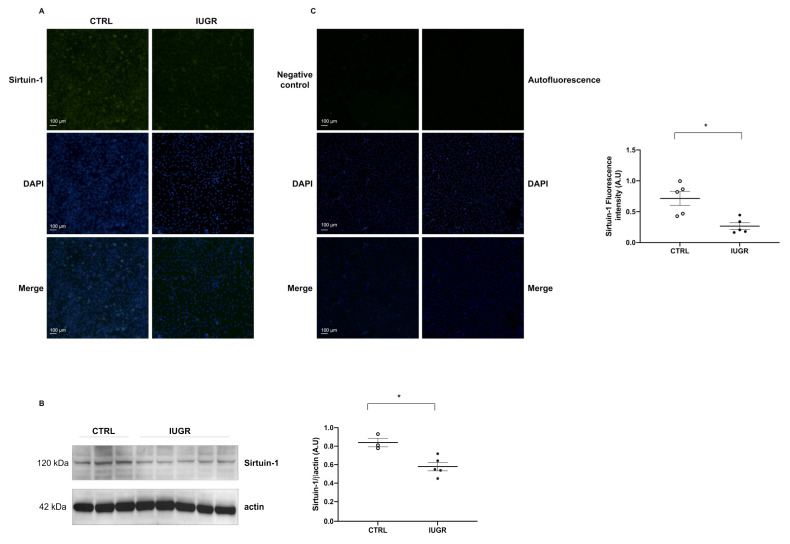
Sirtuin-1 expression. Sirtuin-1 protein expression was measured using immunofluorescence in CTRL-ECFCs and IUGR-ECFCs isolated from six-month-old male rats (**A**). Magnification (20×). Nuclei were counterstained with DAPI, and a negative control (with no primary antibody) and a test of autofluorescence were performed (**C**). These pictures are representative images from *n* = 5 animals/group; * *p* < 0.05. Scale bar = 100 μm. In addition, the sirtuin-1 protein expression was measured using western blot in CTRL-ECFCs and IUGR-ECFCs isolated from six-month-old male rats (**B**); *n* = 3–5 animals/group; * *p* < 0.05.

**Figure 12 ijms-22-10159-f012:**
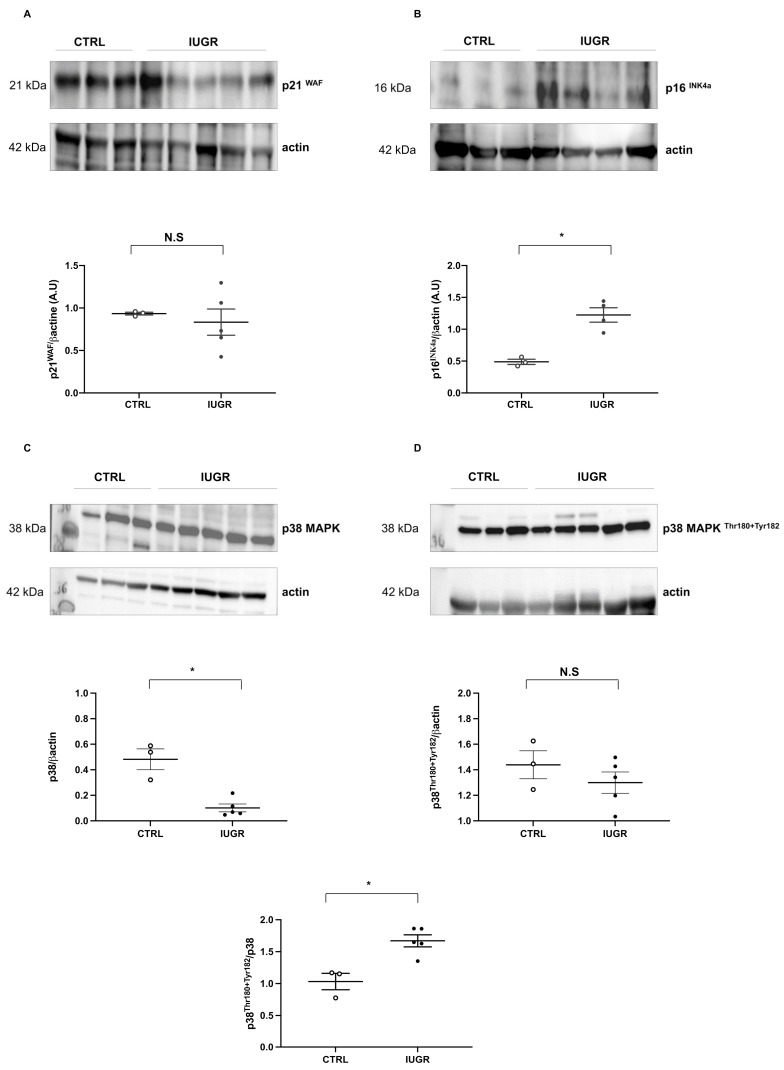
Factors related to cellular senescence. p21^WAF^ (**A**), p16^INK4a^ (**B**), p38 MAPK (**C**) and phosphorylated p38 MAPK^Thr180+Tyr 182^ (**D**) protein content were measured in CTRL-ECFCs and IUGR-ECFCs isolated from six-month-old male rats; *n* = 3–5 animals/group; * *p* < 0.05; N.S: not significant.

**Figure 13 ijms-22-10159-f013:**
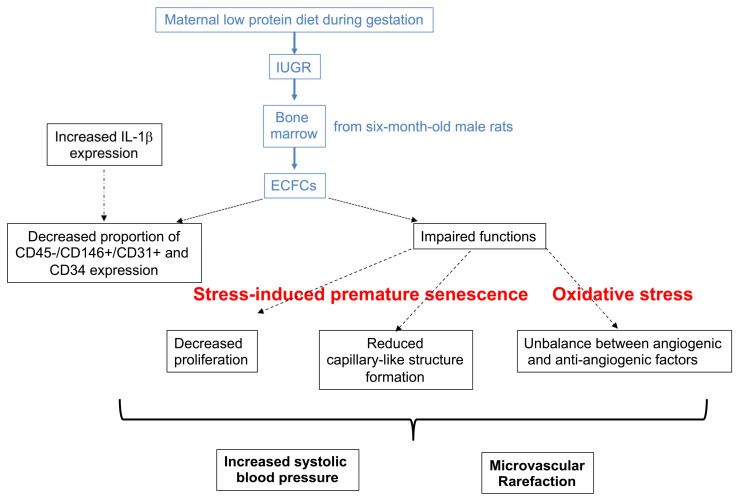
ECFC dysfunction related to IUGR and developmental programming of arterial hypertension at 6 months of life.

**Table 1 ijms-22-10159-t001:** Body weight at birth and at 6 months of life, and systolic blood pressure (SBP) at 6 months of age in CTRL and IUGR males and females.

**Body Weight at Birth**	**CTRL (gram)**	**IUGR (gram)**	**SIGNIFICANCE**
Males (*n* = 25; 5 litters)	7.73 ± 1.03	5.23 ± 0.48	*p* < 0.001
Females (*n* = 25; 5 litters)	7.15 ± 0.48	4.76 ± 0.40	*p* < 0.001
**Body Weight at 6 Months**	**CTRL (gram)**	**IUGR (gram)**	**SIGNIFICANCE**
Males (*n* = 25; 5 litters)	751.81 ± 64.64	586.16 ± 45.43	*p* < 0.001
Females (*n* = 25; 5 litters)	378.62 ± 35.95	319.78 ± 16.94	*p* < 0.001
**SBP at 6 Months of Life**	**CTRL (mmHg)**	**IUGR (mmHg)**	**SIGNIFICANCE**
Males (*n* = 5; 5 litters)	125.72 ± 4.62	153.44 ± 2.51	*p* < 0.01
Females (*n* = 5; 5 litters)	112.56 ± 7.73	113.8 ± 5.92	*p* > 0.05

## Data Availability

https://doi.org/10.5281/zenodo.5512925 accessed on 16 September 2021.

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
