# Peer review of "Endothelial Colony-Forming Cells Dysfunctions Are Associated with Arterial Hypertension in a Rat Model of Intrauterine Growth Restriction"

_ijms, 2021, doi:10.3390/ijms221810159_

Round 1

Reviewer 1 Report

Well presented and well designed study. I have a question, what will be the future directions and how the assesment of endothelial cells could be used in humans in order to predict and prevent preeclampsia and IUGR phenomenon ?

Author Response

We would like to thank the reviewers for their valuable remarks. Our manuscript has been carefully revised according to their requests and suggestions.

Please find below our point-by-point responses (in blue) to the reviewers’ comments (in black).

Reviewer's comments:

Well presented and well designed study.

I have a question, what will be the future directions and how the assessment of endothelial cells could be used in humans in order to predict and prevent preeclampsia and IUGR phenomenon ?

Authors’ response:

We thank the reviewer for his/her relevant suggestions. The section ‘Perspectives’ has been completed in the revised manuscript to better describe future directions.

Modifications in the revised manuscript:

Perspectives

In this study the identification of mechanisms related to ECFC dysfunction, such as oxidative stress and SIPS, could enable us to design specific therapeutic or preventive strategies and to accelerate the research for vascular regenerative therapies. Particularly, it would be interesting to explore whether an antioxidant therapy could restore the functional properties of IUGR-ECFCs, such as proliferation, capillary-like structure formation and expression of angiogenic factors, associated with a decrease in oxidative stress and reversion of SIPS. Resveratrol is widely known as a phenolic compound with powerful antioxidant activity. Resveratrol is present in several plants, including grape skins, grape seeds, giant knotweed, cassia seeds, passion fruit, white tea, plums and peanuts [1, 2]. Wang et al. demonstrated that resveratrol promotes the proliferation, adhesion, migration of EPCs in a dose- and time-dependent manner and increases the expression of VEGF to further induce vasculogenesis [3, 4], which was mediated by the activation of sirtuin-1 [5]. Resveratrol also delays the senescence of EPCs by increasing telomerase activity to maintain the appropriate levels and function of EPCs [6, 7], and by increasing sirtuin-1 functionality [8]. Resveratrol also prevents oxidative stress induced by diabetes in EPCs via sirtuin-1 activation [9]. In ECFCs isolated from low-birth-weight newborns, in vitro treatment with resveratrol has improved ECFC functionality and reversed SIPS; however, whether resveratrol could exert similar actions on ECFCs-IUGR isolated 6 months after birth is still unknown.

In addition, as mentioned above, it will be interesting to explore if impaired ECFC functionalities precede, or are rather a consequence of arterial hypertension by exploring the functionality of ECFCs at birth and at a younger age when SBP is not increased. In addition, although females did not have increased SBP at six months of life, it will be interesting to investigate their ECFC functionality. If ECFC alterations are observed in 6-month-old females, it would be interesting to study whether SBP increases later in life.

Further investigation of epigenetic processes implicated in the regulation of molecular mechanisms identified in this study could be of interest to better understand the developmental programming of hypertension after IUGR.

Because individuals born after IUGR may have subsequent catch-up growth that can amplify cardiometabolic disease, it would also be interesting to observe whether a growth catch-up (induced by litter size restriction during the lactation period) could amplify the adverse effects related to the IUGR in the present rat model.

Finally, the use of stem cells has emerged as promising for regenerative medicine because of their capacity to contribute to organ repair and regeneration throughout life. In particular, EPCs have been identified as having clinical potential, not only in vascular regenerative applications [10, 11] in ischemic diseases such as myocardial infarction and peripheral vascular disease, but also in metabolic diseases and pulmonary and systemic hypertension [12, 13] . In particular, ECFCs represent ideal stem cell candidates thanks to their properties of proliferation, autorenewal, migration, differentiation, vascular growth and neovascularization [14]. Indeed, intrajugular administration of human cord blood-derived ECFCs in newborn rodents was able to reverse alveolar growth arrest, preserve lung vascularity and reduce pulmonary hypertension in a model of hyperoxia-induced bronchopulmonary dysplasia [15]. This cell therapy also prevented cardiomyocyte hypertrophy, as well as the myocardial and perivascular fibrosis observed after neonatal hyperoxia exposure [16].

Concerning clinical applications, ECFCs could provide an interesting tool in the management of preeclampsia and IUGR and their adverse consequences. Whether ECFC dysfunctions are already present at birth, they could be used as biomarkers to identify individuals with an increased risk to develop cardiometabolic disease later in life and to design specific follow-up or preventative approaches for such individuals. Moreover, identification of mechanisms implicated in ECFC dysfunctions could help to design potential treatment to reverse these alterations, as mentioned above. Such an approach could enable treatment with ECFCs isolated from cord blood before re-injection in the neonate to limit long-term adverse effects of IUGR or preeclampsia. Finally, identification of ECFC dysfunctions in maternal blood in pregnancies complicated by IUGR or preeclampsia could be useful as an early diagnostic tool to predict such complications and to improve their management. This could facilitate the design of therapeutic interventions to limit or prevent the development of IUGR or preeclampsia and thus prevent or limit their adverse consequences. Indeed preeclampsia often results in IUGR or preterm babies. The level of circulating ECFCs in cord blood of preeclamptic pregnancies was reduced [17-19] and impaired angiogenic factors have been associated with preeclampsia. Notably, the angiogenic factor VEGF plays a major role in the management of blood pressure during preeclampsia, and low levels of VEGF have been observed in preeclampsia [20]. Exogenous administration of VEGF has been shown to reverse the antiangiogenic effects of preeclamptic plasma [21], and VEGF represents an important regulator of ECFC functionality. So based on our present study and these independent observations, future experiments could focus on the “rescue” of ECFC functionality, either by pharmacological treatment or gene therapy, notably by increasing their angiogenic potential by in vitro conditioning (eNOS, VEGF, CD146) as previously published [22, 23].

References

  1. Piotrowska, H.; Kucinska, M.; Murias, M., Biological activity of piceatannol: leaving the shadow of resveratrol. Mutat Res 2012, 750, (1), 60-82.
  2. Mei, Y. Z.; Liu, R. X.; Wang, D. P.; Wang, X.; Dai, C. C., Biocatalysis and biotransformation of resveratrol in microorganisms. Biotechnol Lett 2015, 37, (1), 9-18.
  3. Wang, X. B.; Huang, J.; Zou, J. G.; Su, E. B.; Shan, Q. J.; Yang, Z. J.; Cao, K. J., Effects of resveratrol on number and activity of endothelial progenitor cells from human peripheral blood. Clin Exp Pharmacol Physiol 2007, 34, (11), 1109-15.
  4. Wallerath, T.; Deckert, G.; Ternes, T.; Anderson, H.; Li, H.; Witte, K.; Forstermann, U., Resveratrol, a polyphenolic phytoalexin present in red wine, enhances expression and activity of endothelial nitric oxide synthase. Circulation 2002, 106, (13), 1652-8.
  5. Gracia-Sancho, J.; Villarreal, G., Jr.; Zhang, Y.; Garcia-Cardena, G., Activation of SIRT1 by resveratrol induces KLF2 expression conferring an endothelial vasoprotective phenotype. Cardiovasc Res 2010, 85, (3), 514-9.
  6. Xia, L.; Wang, X. X.; Hu, X. S.; Guo, X. G.; Shang, Y. P.; Chen, H. J.; Zeng, C. L.; Zhang, F. R.; Chen, J. Z., Resveratrol reduces endothelial progenitor cells senescence through augmentation of telomerase activity by Akt-dependent mechanisms. Br J Pharmacol 2008, 155, (3), 387-94.
  7. Wang, X. B.; Zhu, L.; Huang, J.; Yin, Y. G.; Kong, X. Q.; Rong, Q. F.; Shi, A. W.; Cao, K. J., Resveratrol-induced augmentation of telomerase activity delays senescence of endothelial progenitor cells. Chin Med J (Engl) 2011, 124, (24), 4310-5.
  8. Vassallo, P. F.; Simoncini, S.; Ligi, I.; Chateau, A. L.; Bachelier, R.; Robert, S.; Morere, J.; Fernandez, S.; Guillet, B.; Marcelli, M.; Tellier, E.; Pascal, A.; Simeoni, U.; Anfosso, F.; Magdinier, F.; Dignat-George, F.; Sabatier, F., Accelerated senescence of cord blood endothelial progenitor cells in premature neonates is driven by SIRT1 decreased expression. Blood 2014.
  9. Wu, H.; Li, G. N.; Xie, J.; Li, R.; Chen, Q. H.; Chen, J. Z.; Wei, Z. H.; Kang, L. N.; Xu, B., Resveratrol ameliorates myocardial fibrosis by inhibiting ROS/ERK/TGF-beta/periostin pathway in STZ-induced diabetic mice. BMC Cardiovasc Disord 2016, 16, 5.
  10. Basile, D. P.; Yoder, M. C., Circulating and tissue resident endothelial progenitor cells. J Cell Physiol 2014, 229, (1), 10-6.
  11. Yoder, M. C., Endothelial progenitor cell: a blood cell by many other names may serve similar functions. J Mol Med (Berl) 2013, 91, (3), 285-95.
  12. Chong, M. S.; Ng, W. K.; Chan, J. K., Concise Review: Endothelial Progenitor Cells in Regenerative Medicine: Applications and Challenges. Stem Cells Transl Med 2016, 5, (4), 530-8.
  13. Wang, X. X.; Zhang, F. R.; Shang, Y. P.; Zhu, J. H.; Xie, X. D.; Tao, Q. M.; Zhu, J. H.; Chen, J. Z., Transplantation of autologous endothelial progenitor cells may be beneficial in patients with idiopathic pulmonary arterial hypertension: a pilot randomized controlled trial. J Am Coll Cardiol 2007, 49, (14), 1566-71.
  14. Ambasta, R. K.; Kohli, H.; Kumar, P., Multiple therapeutic effect of endothelial progenitor cell regulated by drugs in diabetes and diabetes related disorder. J Transl Med 2017, 15, (1), 185.
  15. Alphonse, R. S.; Vadivel, A.; Fung, M.; Shelley, W. C.; Critser, P. J.; Ionescu, L.; O'Reilly, M.; Ohls, R. K.; McConaghy, S.; Eaton, F.; Zhong, S.; Yoder, M.; Thebaud, B., Existence, functional impairment, and lung repair potential of endothelial colony-forming cells in oxygen-induced arrested alveolar growth. Circulation 2014, 129, (21), 2144-57.
  16. Girard-Bock, C.; de Araujo, C. C.; Bertagnolli, M.; Mai-Vo, T. A.; Vadivel, A.; Alphonse, R. S.; Zhong, S.; Cloutier, A.; Sutherland, M. R.; Thebaud, B.; Nuyt, A. M., Endothelial colony-forming cell therapy for heart morphological changes after neonatal high oxygen exposure in rats, a model of complications of prematurity. Physiol Rep 2018, 6, (22), e13922.
  17. Munoz-Hernandez, R.; Miranda, M. L.; Stiefel, P.; Lin, R. Z.; Praena-Fernandez, J. M.; Dominguez-Simeon, M. J.; Villar, J.; Moreno-Luna, R.; Melero-Martin, J. M., Decreased level of cord blood circulating endothelial colony-forming cells in preeclampsia. Hypertension 2014,64, (1), 165-71.
  18. Luppi, P.; Powers, R. W.; Verma, V.; Edmunds, L.; Plymire, D.; Hubel, C. A., Maternal circulating CD34+VEGFR-2+ and CD133+VEGFR-2+ progenitor cells increase during normal pregnancy but are reduced in women with preeclampsia. Reprod Sci 2010, 17, (7), 643-52.
  19. Monga, R.; Buck, S.; Sharma, P.; Thomas, R.; Chouthai, N. S., Effect of preeclampsia and intrauterine growth restriction on endothelial progenitor cells in human umbilical cord blood. J Matern Fetal Neonatal Med 2012, 25, (11), 2385-9.
  20. Rana, S.; Karumanchi, S. A.; Lindheimer, M. D., Angiogenic factors in diagnosis, management, and research in preeclampsia. Hypertension 2014, 63, (2), 198-202.
  21. Karumanchi, S. A., Angiogenic Factors in Preeclampsia: From Diagnosis to Therapy. Hypertension 2016, 67, (6), 1072-9.
  22. Smadja, D. M.; Cornet, A.; Emmerich, J.; Aiach, M.; Gaussem, P., Endothelial progenitor cells: characterization, in vitro expansion, and prospects for autologous cell therapy. Cell Biol Toxicol 2007, 23, (4), 223-39.
  23. Essaadi, A.; Nollet, M.; Moyon, A.; Stalin, J.; Simoncini, S.; Balasse, L.; Bertaud, A.; Bachelier, R.; Leroyer, A. S.; Sarlon, G.; Guillet, B.; Dignat-George, F.; Bardin, N.; Blot-Chabaud, M., Stem cell properties of peripheral blood endothelial progenitors are stimulated by soluble CD146 via miR-21: potential use in autologous cell therapy. Sci Rep 2018, 8, (1), 9387.

Reviewer 2 Report

Authors present a work addressing: ”Endothelial colony forming cells dysfunctions are associated with arterial hypertension in a rat model of intrauterine growth restriction”. The general conclusion demonstrates that an impaired functionality of ECFCs at adulthood associated with arterial hypertension in individuals born after intrauterine growth restriction. The topic of the article is relevant for clinical practice. However, the paper presents a few minor issues including:

  1. The authors should redesign the figure 2, 6 and 11 since are completely unreadable.
  2. Generally figures are slightly fuzzy.
  3. The authors should provide limitations of the study.
  4. The manuscript should be checked in respect to punctuation.

Author Response

We would like to thank the reviewers for their valuable remarks. Our manuscript has been carefully revised according to their requests and suggestions.

Please find below our point-by-point responses (in blue) to the reviewers’ comments (in black).

Reviewer's comments:

Authors present a work addressing: “Endothelial colony forming cells dysfunctions are associated with arterial hypertension in a rat model of intrauterine growth restriction”. The general conclusion demonstrates that an impaired functionality of ECFCs at adulthood associated with arterial hypertension in individuals born after intrauterine growth restriction. The topic of the article is relevant for clinical practice. However, the paper presents a few minor issues including:

1- The authors should redesign the figure 2, 6 and 11 since are completely unreadable.

Authors’ response:

The figure 2, 6 and 11 have been redesigned. The initial figure 11 has been split in figure 11 and figure 12 to improve the reading.

In addition, all the figures of this manuscript have been improved, with increasing the text size.

2- Generally figures are slightly fuzzy.

Authors’ response:

The size of the figures has been re-adapted to the size of manuscript to limit that the fuzzy appearance.

3- The authors should provide limitations of the study.

Authors’ response:

We apologize for forgetting to discuss limitations. A section ‘Limitations’ has been included in the revised manuscript.

Modifications in the revised manuscript:

Limitations

The present study was performed only in six-month-old male rats. Therefore, it was not possible to determine whether the observed ECFC alterations precede the increase in SBP. To answer this question it will be necessary to explore the functionality of ECFCs at a younger age at which SBP is not increased. In addition, ECFCs isolated from six-month-old females have not been investigated in this study because of the absence of an increase in SBP in these individuals. It will be therefore necessary to determine whether ECFCs are also altered in females even in the absence of increased SBP.

Finally, the present data were obtained in a rat model of IUGR induced by a maternal low-protein diet. Further investigation should be therefore performed in humans in order to determine whether similar alterations could be observed.

4- The manuscript should be checked in respect to punctuation.

Authors’ response:

The revised manuscript has been checked in respect to punctuation by one of the co-authors, Dr Anne Wilson, who is an English speaker.